# EFFECTIVE CROSS-INSTANCE POSITIVE RELATIONS FOR GENERALIZED CATEGORY DISCOVERY

## ABSTRACT

We tackle the issue of generalized category discovery (GCD). GCD considers the open-world problem of automatically clustering a partially labelled dataset, in which the unlabelled data contain instances from novel categories and also the labelled classes. In this paper, we address the GCD problem without a known category number in the unlabelled data. We propose a framework, named CiP, to bootstrap the representation by exploiting **C**ross-**i**nstance **P**ositive relations for contrastive learning in the partially labelled data which are neglected in existing methods. First, to obtain reliable cross-instance relations to facilitate the representation learning, we introduce a semi-supervised hierarchical clustering algorithm, named selective neighbor clustering (SNC), which can produce a clustering hierarchy directly from the connected components in the graph constructed by selective neighbors. We also extend SNC to be capable of label assignment for the unlabelled instances with the given class number. Moreover, we present a method to estimate the unknown class number using SNC with a joint reference score considering clustering indexes of both labelled and unlabelled data. Finally, we thoroughly evaluate our CiP framework on public generic image recognition datasets (CIFAR-10, CIFAR-100, and ImageNet-100) and challenging fine-grained datasets (CUB, Stanford Cars, and Herbarium19), all establishing the new state-of-the-art.

## 1 INTRODUCTION

After training on large-scale datasets with human annotations, existing machine learning models can achieve superb performance (*e.g.*, (Krizhevsky et al., 2012)). However, the success of these models heavily relies on the fact that they are only tasked to recognize images from the same set of classes with large-scale human annotations on which they are trained. This limits their application in the real open world where we will encounter data without annotations and from unseen categories. Indeed, more and more efforts have been devoted to dealing with more realistic settings. For example, semi-supervised learning (SSL) (Chapelle et al., 2006) aims at training a robust model using both labelled and unlabelled data from the same set of classes; few-shot learning (Snell et al., 2017) tries to learn models that can generalize to new classes with few annotated samples; open-set recognition (OSR) (Scheirer et al., 2012) learns to tell whether or not an unlabelled image belongs to one of the classes on which the model is trained. More recently, the problem of novel category discovery (NCD) (Han et al., 2019; 2020; Fini et al., 2021) has been introduced, which learns models to automatically partition unlabelled data from unseen categories by transferring knowledge from seen categories. One assumption in early NCD methods is that unlabelled images are all from unseen categories only. NCD has been recently extended to a more generalized setting, called generalized category discovery (GCD) (Vaze et al., 2022b), by relaxing the assumption to reflect the real world better, *i.e.*, unlabelled images are from both seen and unseen categories.

In this paper, we tackle the problem of GCD by drawing inspiration from the baseline method (Vaze et al., 2022b). In (Vaze et al., 2022b), a vision transformer model was first trained for representation learning using supervised contrastive learning on labelled data and self-supervised contrastive learning on both labelled and unlabelled data. With the learned representation, semi-supervised $k$-means (Han et al., 2019) was then adopted for label assignment across all instances. In addition, based on semi-supervised $k$-means, (Vaze et al., 2022b) also introduced an algorithm to estimate the unknown category number for the unlabelled data by examining possible category numbers in a given range. However, this approach has several limitations. First, during representation learning, the method considers labelled and unlabelled data independently, and uses a stronger training signal for the

labelled data which might compromise the representation of the unlabelled data. Second, the method requires a known category number for performing label assignment. Third, the category number estimation method is slow as it needs to run the clustering algorithm multiple times to test different category numbers.

To overcome the above limitations, we propose a new approach for GCD which does not require a known unseen category number and considers **C**ross-**i**nstance **P**ositive relations in unlabelled data for better representation learning (CiP). At the core of our approach is our novel semi-supervised hierarchical clustering algorithm with selective neighbor, named as selective neighbor clustering (SNC), that takes inspiration from the parameter-free hierarchical clustering method FINCH (Sarfraz et al., 2019). SNC can not only generate reliable pseudo labels for cross-instance positive relations, but also estimate unseen category numbers without the need for repeated runs of the clustering algorithm. SNC builds a graph indicating all subtly selected neighbor relations constrained by the labelled instances, and produces clusters directly from the connected components in the graph. SNC iteratively constructs a hierarchy of partitions with different granularity, while satisfying the constraints imposed by the labelled instances. With a one-by-one merging strategy, SNC can quickly estimate a reliable class number without repeated runs of the algorithm, which makes it significantly faster than (Vaze et al., 2022b).

The main contributions of this paper can be summarized as follows: (1) we propose a new GCD framework, named CiP, exploiting more cross-instance positive relations in the partially labelled set to strengthen the connections among all instances, fostering the representation learning for better category discovery; (2) we introduce a semi-supervised hierarchical clustering algorithm, named SNC, that can be adopted for reliable pseudo label generation during training and label assignment during testing; (3) we further leverage SNC for class number estimation by exploring intrinsic and extrinsic clustering quality based on a joint reference score considering both labelled and unlabelled data; (4) we comprehensively evaluate our CiP framework on both generic image recognition datasets and challenging fine-grained datasets, and demonstrate state-of-the-art performance across the board.

## 2 RELATED WORK

Our work is related to novel/generalized category discovery, semi-supervised learning, and open-set recognition.

*Novel category discovery (NCD)* aims at discovering new classes in unlabelled data by leveraging knowledge learned from labelled data. It was pioneered by (Han et al., 2019) with a transfer clustering approach. Some earlier works on cross-domain/task transfer learning (Hsu et al., 2018a;b) can also be adopted to tackle this problem. (Han et al., 2020) proposed an efficient method called AutoNovel (aka RankStats) using ranking statistics. They first learned a good embedding using low-level self-supervised learning on all data followed by supervised learning on labelled data for higher level features. They introduced a robust ranking statistics to determine whether two unlabelled instances are from the same class for NCD. Several successive works based on RankStats were proposed. For example, (Jia et al., 2021) proposed to use WTA hashing (Yagnik et al., 2011) for NCD in single- and multi-modal data; Zhao and Han (Zhao & Han, 2021) extended NCD with dual ranking statistics and knowledge distillation. (Fini et al., 2021) proposed UNO which uses a unified cross entropy loss to train labelled and unlabelled data. (Chi et al., 2022) proposed meta discovery which links NCD to meta learning with limited labelled data. (Vaze et al., 2022b) introduced generalized category discovery (GCD) which extends NCD by allowing unlabelled data from both old and new classes. They first finetuned a pretrained DINO ViT (Caron et al., 2021) with both supervised contrastive loss and self-supervised contrastive loss. Semi-supervised $k$-means was then adopted for label assignment. A concurrent work called ORCA by (Cao et al., 2022) addressed a similar problem by formulating it as open-world semi-supervised learning. We draw inspiration from (Vaze et al., 2022b) and develop a novel method to tackle GCD by exploring cross-instance correlations on labelled and unlabelled data which have been neglected in (Vaze et al., 2022b).

*Semi-supervised learning (SSL)* has long been studied in the machine learning community (Chapelle et al., 2006). It aims at learning a good model by leveraging unlabelled data from the same set of classes as the labelled data. Various methods have been proposed for SSL. For example, $\Pi$-model (Laine & Aila, 2017) uses self-ensembling to leverage label predictions on different epochs and under different conditions; Mean Teacher (Tarvainen & Valpola, 2017) utilizes averaging model weights instead of label predictions; FixMatch (Sohn et al., 2020) and FlexMatch (Zhang et al., 2021)

employ pseudo-labels generated from model predictions to guide the training. The assumption that labelled and unlabelled data are from the same closed set of classes is often not valid in practice. In contrast, GCD relaxes this assumption and considers a more challenging scenario where unlabelled data can also come from unseen classes.

*Open-set recognition (OSR)* aims at training a model using data from a known closed set of classes, and at test time determining whether or not a sample is from one of these known classes. It was first introduced in (Scheirer et al., 2012). Since then many methods have been proposed for this task. For example, OpenMax (Bendale & Boult, 2016) is the first deep learning work to address the OSR problem based on Extreme Value Theory and fitting per-class Weibull distributions. RPL (Chen et al., 2020a) and its extension ARPL (Chen et al., 2021) exploit reciprocal points for constructing extra-class space to reduce the risk of unknown. Recently, (Vaze et al., 2022a) found the correlation between closed and open-set performance, and boosted the performance of OSR by improving closed-set accuracy. They also proposed Semantic Shift Benchmark (SSB) with a clear definition of semantic novelty for better OSR evaluation.

## 3 METHODOLOGY

### 3.1 PROBLEM FORMULATION

Generalized category discovery (GCD) aims at automatically categorizing unlabelled images in a collection of data in which part of the data is labelled and the rest is unlabelled. The unlabelled images may come from the labelled classes or new ones. This is a much more realistic open-world setting than the common closed-set classification where the labelled and unlabelled data are from the same set of classes. Let the data collection be $\mathcal{D} = \mathcal{D}_\mathcal{L} \cup \mathcal{D}_\mathcal{U}$, where $\mathcal{D}_\mathcal{L} = \{(x_i^\ell, y_i^\ell)\}_{i=1}^M \in \mathcal{X} \times \mathcal{Y}_\mathcal{L}$ denotes the labelled subset and $\mathcal{D}_\mathcal{U} = \{(x_i^u, y_i^u)\}_{i=1}^N \in \mathcal{X} \times \mathcal{Y}_\mathcal{U}$ denotes the unlabelled subset with unknown $y_i^u \in \mathcal{Y}_\mathcal{U}$. Only a subset of classes contains labelled instances, *i.e.*, $\mathcal{Y}_\mathcal{L} \subset \mathcal{Y}_\mathcal{U}$. The number of labelled classes $N_\mathcal{L}$ can be directly deduced from the labelled data, while the number of unlabelled classes $N_\mathcal{U}$ is not known a priori.

To tackle this challenge, we propose a novel framework CiP to jointly learn representations using contrastive learning by considering all possible interactions between labelled and unlabelled instances. Contrastive learning has been applied to learn representation in GCD, but without considering the connections between labelled and unlabelled instances (Vaze et al., 2022b) due to the lack of reliable pseudo labels. This limits the learned representation. In this paper, we propose an efficient semi-supervised hierarchical clustering algorithm, named selective neighbor clustering (SNC), to generate reliable pseudo labels to bridge labelled and unlabelled instances during training and bootstrap representation learning. With the generated pseudo labels, we can then train the model on both labelled and unlabelled data in a supervised manner considering all possible pairwise connections. We further extend SNC with a simple one-by-one merging process to allow cluster number estimation and label assignment on all unlabelled instances. An overview of our CiP is shown in Fig. 1.

### 3.2 JOINT CONTRASTIVE REPRESENTATION LEARNING

Contrastive learning has been widely used for self-supervised representation learning (Chen et al., 2020b; He et al., 2020) and supervised representation learning (Khosla et al., 2020). For GCD, since the data contains both labelled and unlabelled instances, the mix of self-supervised and supervised contrastive learning appears to be a natural fit and good performance has been reported in (Vaze et al., 2022b). However, cross-instance correlations are only considered for pairs of labelled instances, but not for pairs of unlabelled instances and pairs of labelled and unlabelled instances. The learned representation is likely to be biased towards the labelled data due to the stronger learning signal provided by them. Meanwhile, the embedding spaces learned from cross-instance correlations of labelled data and self correlations of unlabelled data might not be necessarily well aligned. These might explain why a much stronger performance on labelled data was reported in (Vaze et al., 2022b) compared with the unlabelled data. To mediate such a bias, we propose to introduce cross-instance relations for pairs of unlabelled instances and pairs of labelled and unlabelled instances in contrastive learning to bootstrap the representation learning. To this end, we propose an efficient semi-supervised hierarchical clustering algorithm to generate reliable pseudo labels relating pairs of unlabelled instances and pairs of labelled and unlabelled instances, as will be detailed in Sec. 3.3. Next, we briefly review supervised contrastive learning (Khosla et al., 2020), which accommodates cross-instance relations, and describe how to extend it to unlabelled data.

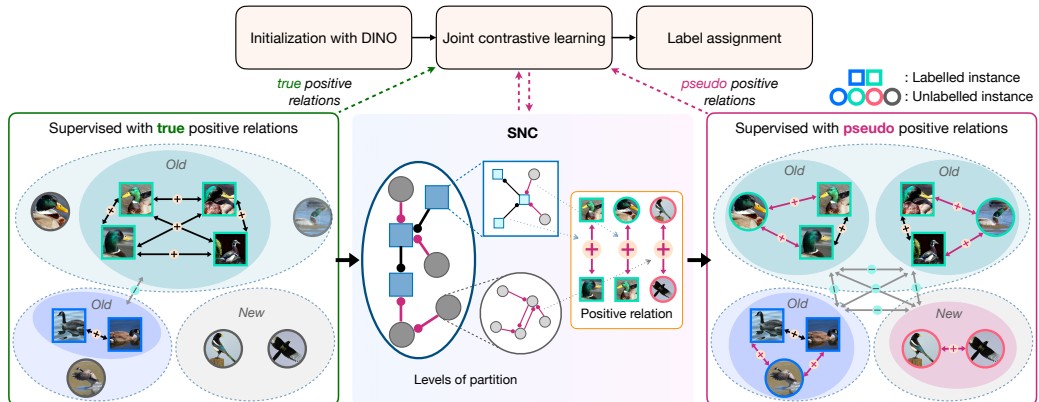

Figure 1: **Overview of our CiP framework.** We first initialize ViT with pretrained DINO (Caron et al., 2021) to obtain a good representation space. We then finetune ViT by conducting joint contrastive learning with both true and pseudo positive relations in a supervised manner. True positive relations come from labelled data while pseudo positive relations of all data are generated by our proposed SNC algorithm. Specifically, SNC generates a hierarchical clustering structure. Pseudo positive relations are granted to all instances in the same cluster at one level of partition, further exploited in joint contrastive learning. With representations well learned, we estimate class number and assign labels to all unlabelled data using SNC with a one-by-one merging strategy.

Let $f$ and $\phi$ be a feature extractor and a MLP projection head. The supervised contrastive loss on labelled data can be formulated as

$$\mathcal{L}_i^s = -\frac{1}{|\mathcal{G}_\mathcal{B}(i)|} \sum_{q \in \mathcal{G}_\mathcal{B}(i)} \log \frac{\exp(\boldsymbol{z}_i^\ell \cdot \boldsymbol{z}_q^\ell / \tau_s)}{\sum_{n \in \mathcal{B}_\mathcal{L}, n \neq i} \exp(\boldsymbol{z}_i^\ell \cdot \boldsymbol{z}_n^\ell / \tau_s)} \tag{1}$$

where $\boldsymbol{z}^\ell = \phi(f(x^\ell))$, $\tau_s$ is the temperature, and $\mathcal{G}_\mathcal{B}(i)$ denotes other instances sharing the same label with the $i$-th labelled instance in $\mathcal{B}_\mathcal{L}$, which is the labelled subset in the mini-batch $\mathcal{B}$. Supervised contrastive loss leverages the true cross-instance positive relations between labelled instance pairs. To take into account the cross-instance positive relations for pairs of unlabelled instances and pairs of labelled and unlabelled instances, we extend the supervised contrastive loss on *all* data as

$$\mathcal{L}_i^a = -\frac{1}{|\mathcal{P}_\mathcal{B}(i)|} \sum_{q \in \mathcal{P}_\mathcal{B}(i)} \log \frac{\exp(\boldsymbol{z}_i \cdot \boldsymbol{z}_q / \tau_a)}{\sum_{n \in \mathcal{B}, n \neq i} \exp(\boldsymbol{z}_i \cdot \boldsymbol{z}_n / \tau_a)} \tag{2}$$

where $\tau_a$ is the temperature, $\mathcal{P}_\mathcal{B}(i)$ is the set of pseudo positive instances for the $i$-th instance in the mini-batch $\mathcal{B}$. The overall loss considering cross-instance relations for pairs of labelled instances, unlabelled instances, as well as labelled and unlabelled instances can then be written as

$$\mathcal{L} = \sum_{i \in \mathcal{B}} \mathcal{L}_i^a + \sum_{i \in \mathcal{B}_\mathcal{L}} \mathcal{L}_i^s \tag{3}$$

With the learned representation, we can discover classes with existing algorithms like semi-supervised $k$-means (Han et al., 2019; Vaze et al., 2022b). We further propose a new method in Sec. 3.4 based on our pseudo label generation approach as will be introduced next.

### 3.3 SELECTIVE NEIGHBOR CLUSTERING

To generate pseudo labels for Eq. (2), an intuitive approach would be to apply an off-the-shelf clustering method like $k$-means or semi-supervised $k$-means to construct clusters and then obtain cross-instance relations based on the resulting cluster assignment. However, we empirically found that such a simple approach will produce many false positive pairs which severely hurt the representation learning. One way to tackle this problem is to overcluster the data to lower the false positive rate. FINCH (Sarfraz et al., 2019) has shown superior performance on unsupervised hierarchical overclustering, but it is non-trivial to extend it to cover both labelled and unlabelled data. Experiments show that FINCH will fail drastically if we simply include all the labelled data. Inspired by FINCH, we propose an efficient semi-supervised hierarchical clustering algorithm, named SNC, with selective neighbor, which subtly makes use of the labelled instances during clustering.

FINCH constructs an adjacency matrix $A$ for all possible pairs of instances $(i, j)$, given by

$$A(i,j) = \begin{cases} 1 & \text{if } j = \kappa_i \text{ or } \kappa_j = i \text{ or } \kappa_i = \kappa_j \\ 0 & \text{else} \end{cases}, \tag{4}$$

where $\kappa_i$ is the first neighbor of the $i$-th instance and is defined as

$$\kappa_i = \arg\max_j \{f(x_i) \cdot f(x_j) \mid x_j \in \mathcal{D}\}, \tag{5}$$

where $f(\cdot)$ outputs an $\ell_2$-normalized feature vector. A data partition can then be obtained by extracting connected components from $A$. Each connected component in $A$ corresponds to one cluster. By treating each cluster as a super instance and building the first neighbor adjacency matrix iteratively, the algorithm can produce hierarchical partitions.

First neighbor is designed for purely unlabelled data. To make use of the labels in partially labelled data, a straightforward idea is to connect all labelled data from the same class by setting $A(i, j)$ to 1 for all pairs of instances $(i, j)$ from the same class. However, after filling $A(i, j)$ for pairs of unlabelled instances using Eq. (4), very often all instances become connected to a single cluster, making it impossible to properly partition the data. This problem is caused by having too many links among the labelled instances. To solve this problem, we would like to reduce the links between labelled instances while keeping labelled instances from the same class in the same connected component. A simple idea is to connect same labelled instances one by one to form a *chain*, which can significantly reduce the number of links. However, we found this still produces many incorrect links, resulting in low purity of the connected components. To this

---

**Algorithm 1** Selective Neighbor Clustering (SNC)

1: **Preparation:**
2: Given labelled set $\mathcal{D}_\mathcal{L}$ and unlabelled set $\mathcal{D}_\mathcal{U}$, treat each instance in $\mathcal{D}_\mathcal{L} \cup \mathcal{D}_\mathcal{U}$ as a cluster $\mathbf{c}_i^0$ with the cluster centroid $\mu(\mathbf{c}_i^0)$ being each instance itself, forming the first partition $\Gamma^0 = \Gamma_\mathcal{L}^0 \cup \Gamma_\mathcal{U}^0$, where $\Gamma^0 = \{\mathbf{c}_i^0\}_{i=1}^{|\Gamma_\mathcal{L}^0| + |\Gamma_\mathcal{U}^0|}$.
3: **Main loop:**
4: $p \leftarrow 0$
5: **while** there are more than $N_\mathcal{L}$ clusters in $\Gamma^p$ **do**
6:     Initialize $\Gamma_\mathcal{L}^\star = \Gamma_\mathcal{L}^p$.
7:     **while** there exists $\kappa_i$ of $\mathbf{c}_i^p \in \Gamma_\mathcal{L}^p \cup \Gamma_\mathcal{U}^p$ not specified **do**
8:         **if** $\mathbf{c}_i^p \in \Gamma_\mathcal{L}^p$ **then**
9:             Initialize $\mathcal{Q} = \{\mathbf{c}_i^p\}, \Gamma_\mathcal{L}^\star = \Gamma_\mathcal{L}^\star \setminus \{\mathbf{c}_i^p\}$.
10:             **while** $|\mathcal{Q}| < \lambda$ **do**
11:                 $\kappa_i \leftarrow \arg\max_j \{\mu(\mathbf{c}_i^p) \cdot \mu(\mathbf{c}_j^p) \mid \mathbf{c}_j^p \in \Gamma_\mathcal{L}^\star, y_j^p = y_i^p\}$
12:                 $\Gamma_\mathcal{L}^\star \leftarrow \Gamma_\mathcal{L}^\star \setminus \{\mathbf{c}_{\kappa_i}^p\}$
13:                 $\mathcal{Q} \leftarrow \mathcal{Q} \cup \{\mathbf{c}_{\kappa_i}^p\}$
14:                 $\mathbf{c}_i^p \leftarrow \mathbf{c}_{\kappa_i}^p$
15:             **end while**
16:         **else**
17:             $\kappa_i \leftarrow \arg\max_j \{\mu(\mathbf{c}_i^p) \cdot \mu(\mathbf{c}_j^p) \mid \mathbf{c}_j^p \in \Gamma_\mathcal{L}^p \cup \Gamma_\mathcal{U}^p\}$
18:         **end if**
19:     **end while**
20:     Construct $A$ following Eq. (4) with selective neighbors, forming a new partition $\Gamma^{p+1} = \Gamma_\mathcal{L}^{p+1} \cup \Gamma_\mathcal{U}^{p+1}$.
21:     $p \leftarrow p + 1$
22: **end while**

---

end, we introduce our selective neighbor to improve the purity of clusters while properly incorporating the labelled instances. The key ideas are as follows. First, we limit the chain length to at most $\lambda$. Second, each labelled instance in a chain can only be the selective neighbor of another labelled instance once. Third, the selective neighbor of an unlabelled instance can be a labelled or an unlabelled instance, depending on its actual distances to other instances. Similar to FINCH, we can apply selective neighbor iteratively to produce hierarchical clustering results. We name our method SNC which is summarized in Algo. 1 (lines 7-19 correspond to selective neighbor computation).

For the chain length $\lambda$, we simply set it to the smallest integer great than or equal to the square root of the number of labelled instances $n_\ell$ in each class, *i.e.*, $\lambda = \lceil \sqrt{n_\ell} \rceil$. This is applied to all classes with labelled instances, and at each hierarchy level. A proper chain length can therefore be dynamically determined based on the actual size of the labelled cluster and also the hierarchy level. We analyze different formulations of chain length in Appx. A.1.

SNC produces a hierarchy of data partitions with different granularity. Except the bottom level, where each individual instance is treated as one cluster, every non-bottom level can be used to capture cross-instance relations for the level below, because each instance in the current level represents a cluster of instances in the level below. In principle, we can pick any non-bottom level to generate pseudo labels. To have a higher purity for each cluster, it is beneficial to choose a relatively low level which overclusters the data. Hence, we choose a level that has a cluster number notably larger than the labelled class number (*e.g.*, $2\times$ more). Meanwhile, the level should not be too low as this will provide much fewer useful pair-wise correlations. In our experiment, we simply pick the third level

from the bottom of hierarchy, which consistently shows good performance on all datasets. We discuss on the impact of the picked level in Appx. A.1.

### 3.4 LABEL ASSIGNMENT WITH AN UNKNOWN CLASS NUMBER

Once a good representation is learned, we could then determine the class label assignment for all unlabelled instances. When the class number is known, we can obtain the label assignment by adopting semi-supervised $k$-means like (Han et al., 2019; Vaze et al., 2022b) or directly using our proposed SNC. Since SNC is an hierarchical clustering algorithm and the cluster number in each hierarchy level is determined automatically by the intrinsic correlations of the instances, it might not produce a level of partition with the exact same cluster number as

---

**Algorithm 2** One-by-one merging

1: **Preparation:**
2: Get initial partitions $S = \{\Gamma^p\}_{p=0}$ by SNC and a cluster number range $[N_e, N_o]$. Note that the merging is from $N_o$ to $N_e$ and $N_o > N_e$.
3: **Partition initialization:**
4: Find $\Gamma^t \in S$ satisfying $|\Gamma^t| > N_o$ and $|\Gamma^{t+1}| \leq N_o$.
5: **Merging:**
6: **while** $|\Gamma^t| > N_e$ **do**
7:     $(i, j) \leftarrow \arg\min_{i,j} \{\mu(\mathbf{c}_i^t) \cdot \mu(\mathbf{c}_j^t) \mid \mathbf{c}_i^t, \mathbf{c}_j^t \in \Gamma^t\}$
8:     Merge $\mathbf{c}_i^t$ and $\mathbf{c}_j^t$, forming a new partition $\Gamma^\star$.
9:     Update current partition $\Gamma^t \leftarrow \Gamma^\star$.
10: **end while**
11: **Output**:
12: Obtain a specific partition $\Gamma^t$ of $N_e$ clusters, *i.e.*, $|\Gamma^t| = N_e$.

---

the known class number. We therefore introduce a simple *one-by-one merging* strategy to SNC allowing it to reach a given class number. Specifically, we first identify a level of partition that has the closest cluster number larger than the given class number, and then merge the clusters one by one until the given class number is reached. At each merging step, we simply merge the two closest clusters. The merging process is summarized in Algo. 2. The label assignment can then be retrieved from the final partition.

When the class number is *unknown*, exiting methods based on semi-supervised $k$-means need to first estimate the unknown cluster number before they can produce the label assignment. To estimate the unknown cluster number, (Han et al., 2019) proposed to run semi-supervised $k$-means on all the data while dropping part of the labels for clustering performance validation. Though effective, this algorithm is computational expensive as it needs to run semi-supervised $k$-means on all possible cluster numbers. (Vaze et al., 2022b) proposed an improved method with Brent's optimization (Brent, 1971), which increases the efficiency. With the estimated cluster number, semi-supervised $k$-means is run again on all labelled and unlabelled instances to produce the final label assignment.

In contrast, SNC can directly produce hierarchical cluster assignments without a known class number. For practical use, one can pick any level of assignments based on the required granularity. To obtain more reliable class number estimation, we propose to use a joint reference score considering both labelled and unlabelled data. In particular, we further split the labelled data $\mathcal{D}_{\mathcal{L}}$ into two parts $\mathcal{D}_{\mathcal{L}}^l$ and $\mathcal{D}_{\mathcal{L}}^v$. We then run SNC on the full dataset $\mathcal{D}$ treating $\mathcal{D}_{\mathcal{L}}^l$ as labelled and $\mathcal{D}_{\mathcal{U}} \cup \mathcal{D}_{\mathcal{L}}^v$ as unlabelled. We then jointly measure the unsupervised intrinsic clustering index (such as silhouette score (Rousseeuw, 1987)) on $\mathcal{D}_{\mathcal{U}}$ and the extrinsic clustering accuracy on $\mathcal{D}_{\mathcal{L}}^v$. We obtain a joint reference score $s_c$ by simply multiplying them after min-max scaling to achieve the best overall measurement on the labelled and unlabelled subsets. We then choose the level in SNC hierarchy with the maximum $s_c$.

The cluster number in the chosen level can be regarded as the estimated class number. To achieve more accurate class number estimation, we further leverage the one-by-one merging strategy. Namely, with the chosen level, we apply the one-by-one merging strategy starting from the level below the chosen one to the level above the chosen one. We then identify the merge that gives the best reference score $s_c$ and consider its cluster number as our estimated class number.

Our proposed SNC with the one-by-one merging strategy can carry out class number estimation with one single run of hierarchical clustering, which is significantly more efficient than the methods based on semi-supervised $k$-means (Han et al., 2019; Vaze et al., 2022a).

## 4 EXPERIMENTS

### 4.1 EXPERIMENTAL SETUP

**Data and evaluation metric.** We evaluate our mothod on three generic image classification datasets, namely CIFAR-10 (Krizhevsky et al., 2009), CIFAR-100 (Krizhevsky et al., 2009), and ImageNet-100 (Deng et al., 2009). ImageNet-100 refers to randomly subsampling 100 classes from the ImageNet dataset. We further evaluate on two more challenging fine-grained image classification datasets,

namely Semantic Shift Benchmark (Vaze et al., 2022a) (SSB includes CUB-200 (Wah et al., 2011) and Stanford Cars (Krause et al., 2013)) and long-tailed Herbarium19 (Tan et al., 2019). We follow (Vaze et al., 2022b) to split the original training set of each dataset into labelled and unlabelled parts. We sample a subset of half the classes as seen categories. $50\%$ of instances of each labelled class are drawn to form the labelled set, and all the rest data constitute the unlabeled set. The model takes all images as input and predicts a label assignment for each unlabelled instance. For evaluation, we measure the clustering accuracy by comparing the predicted label assignment with the ground truth, following the protocol of (Vaze et al., 2022b).

**Implementation details.** We follow (Vaze et al., 2022b) to use the ViT-B-16 initialized with pretrained DINO (Caron et al., 2021) as our backbone. The output `[CLS]` token is used as the feature representation. Following the standard practice, we project the representations with a non-linear projection head and use the projected embeddings for contrastive learning. We set the dimension of projected embeddings to 65,536 following (Caron et al., 2021). At training time, we feed two views with random augmentations to the model. We only fine-tune the last block of the vision transformer with an initial learning rate of 0.01 and the head is trained with an initial learning rate of 0.1. All methods are trained for 200 epochs with cosine annealing schedule. For our method, the temperatures of two supervised contrastive losses $\tau_s$ and $\tau_a$ are set to 0.07 and 0.1 respectively. For class number estimation, we set $|\mathcal{D}_{\mathcal{L}}^l|:|\mathcal{D}_{\mathcal{L}}^v| = 8:2$. Our experiments are conducted on RTX 3090 GPUs.

## 4.2 COMPARISON WITH THE STATE-OF-THE-ART

We compare our CiP with four strong GCD baselines: *RankStats+* and *UNO+*, which are adapted from RankStats (Han et al., 2021) and UNO (Fini et al., 2021) that are originally developed for NCD, the state-of-the-art GCD method of (Vaze et al., 2022b), and *ORCA* (Cao et al., 2022) which addresses GCD from a semi-supervised learning perspective. As ORCA uses a different backbone model and data splits, for fair comparison, we retrain ORCA with ViT model using the official code on the same splits here.

In Tab. 1, we compare CiP with others on the generic image recognition datasets. CiP consistently outperforms all others by a significant margin. For example, CiP outperforms the state-of-the-art GCD method of (Vaze et al., 2022b) by 6.2% on CIFAR-10, 10.7% on CIFAR-100, and 6.4% on ImageNet-100 for 'All' classes, and by 9.5% on CIFAR-10, 22.7% on CIFAR-100, and 12.0% on ImageNet-100 for 'Unseen' classes. This demonstrates cross-instance positive relations obtained by SNC are effective to learn better representations for unlabelled data. Due to the fact that a linear classifier is trained on 'Seen' classes, UNO+ shows a strong performance on 'Seen' classes, but its performance on 'Unseen' ones is significantly worse. In contrast, CiP achieves comparably good performance on both 'Seen' and 'Unseen' classes, without biasing to the labelled data.

Table 1: **Results on generic image recognition datasets.**

| | CIFAR-10 | | | CIFAR-100 | | | ImageNet-100 | | |
|---|---|---|---|---|---|---|---|---|---|
| Classes | All | Seen | Unseen | All | Seen | Unseen | All | Seen | Unseen |
| RankStats+ (Han et al., 2021) | 46.8 | 19.2 | 60.5 | 58.2 | 77.6 | 19.3 | 37.1 | 61.6 | 24.8 |
| UNO+ (Fini et al., 2021) | 68.6 | **98.3** | 53.8 | 69.5 | 80.6 | 47.2 | 70.3 | **95.0** | 57.9 |
| ORCA (Cao et al., 2022) | 97.3 | 97.3 | 97.4 | 66.4 | 70.2 | 58.7 | 38.2 | 67.6 | 23.4 |
| Vaze *et al.* (Vaze et al., 2022b) | 91.5 | 97.9 | 88.2 | 70.8 | 77.6 | 57.0 | 74.1 | 89.8 | 66.3 |
| Ours (CiP) | **97.7**±0.1 | 97.5±0.3 | **97.7**±0.2 | **81.5**±1.2 | **82.4**±1.2 | **79.7**±3.2 | **80.5**±1.4 | 84.9±1.1 | **78.3**±2.4 |

In Tab. 2, we further compare our method with others on fine-grained image recognition datasets, in which the difference between different classes are subtle, making it more challenging for GCD. Again, CiP consistently outperforms all other methods for 'All' and 'Unseen' classes. On CUB-200 and SCars, CiP achieves 5.8% and 8.0% improvement over the state-of-the-art for 'All' classes. For the challenging Herbarium19 dataset, which contains many more classes than other datasets and has the extra challenge of long-tailed distribution, CiP still achieves an improvement of 1.4% and 5.6% for 'All' and 'Unseen' classes. Both RankStats+ and UNO+ show a strong bias to the 'Seen' classes.

In Fig. 2, we visualize the t-SNE projection on features generated by DINO (Caron et al., 2021), GCD method of (Vaze et al., 2022b), and our method CiP, performed on CIFAR-10. Both (Vaze et al., 2022b) and our features are more discriminative than DINO features. The method of (Vaze et al., 2022b) captures better representations with more separable clusters, but some seen categories are confounded with unseen categories, *e.g.*, cat with dog and automobile with truck, while CiP features show better cluster boundaries for seen and unseen categories, further validating the quality of our learned representation.

Table 2: **Results on fine-grained image recognition datasets.**

| Classes | CUB-200 | | | SCars | | | Herbarium19 | | |
|---|---|---|---|---|---|---|---|---|---|
| | All | Seen | Unseen | All | Seen | Unseen | All | Seen | Unseen |
| RankStats+ (Han et al., 2021) | 33.3 | 51.6 | 24.2 | 28.3 | 61.8 | 12.1 | 27.9 | **55.8** | 12.8 |
| UNO+ (Fini et al., 2021) | 35.1 | 49.0 | 28.1 | 35.5 | **70.5** | 18.6 | 28.3 | 53.7 | 14.7 |
| ORCA (Cao et al., 2022) | 35.0 | 35.6 | 34.8 | 32.6 | 47.0 | 25.7 | 24.6 | 26.5 | 23.7 |
| Vaze *et al.* (Vaze et al., 2022b) | 51.3 | 56.6 | 48.7 | 39.0 | 57.6 | 29.9 | 35.4 | 51.0 | 27.0 |
| Ours (CiP) | **57.1**±0.4 | **58.7**±1.8 | **55.6**±0.9 | **47.0**±2.3 | 61.5±3.5 | **40.1**±1.9 | **36.8**±0.6 | 45.4±1.9 | **32.6**±0.3 |

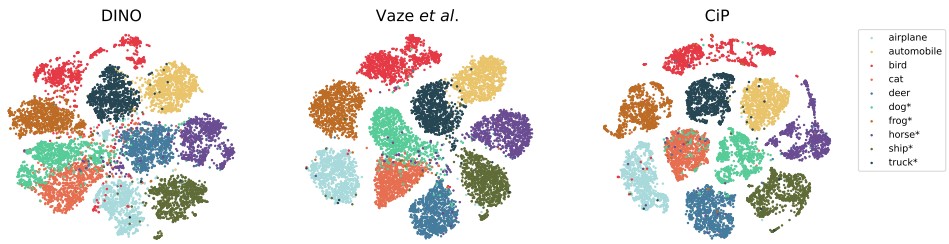

Figure 2: **Visualization on CIFAR-10.** We conduct t-SNE projection on features extracted by raw DINO, GCD method of (Vaze et al., 2022b) and our CiP. We randomly sample 1000 images of each class from CIFAR-10 to visualize. Unseen categories are marked with *.

## 4.3 ESTIMATING THE UNKNOWN CLASS NUMBER

In Tab. 3, we report our estimated class numbers on both generic and fine-grained datasets using the joint reference score $s_c$ as described in Sec. 3.4. Overall, CiP achieves comparable results with the method of (Vaze et al., 2022b) costing slightly more memory, but it is far more efficient (40-150 times faster) and also does not require a list of predefined possible numbers. Even for the most difficult Herbarium19 dataset, CiP only takes a few minutes to finish, while it takes more than an hour for a single run of $k$-means due to large class number, let alone multiple runs from a predefined list of possible class numbers.

Table 3: **Estimation of class number in unlabelled data.**

| | Method | CIFAR-10 | CIFAR-100 | ImageNet-100 | CUB-200 | SCars | Herbarium19 |
|---|---|---|---|---|---|---|---|
| Ground truth | — | 10 | 100 | 100 | 200 | 196 | 683 |
| Estimate (error) | Vaze *et al.* (Vaze et al., 2022b) | 9 (10%) | 100 (0%) | 109 (9%) | 231 (16%) | 230 (17%) | 520 (24%) |
| | Ours (CiP) | 12 (20%) | 103 (3%) | 100 (0%) | 155 (23%) | 182 (7%) | 490 (28%) |
| Runtime | Vaze *et al.* (Vaze et al., 2022b) | 15394s | 27755s | 64524s | 7197s | 8863s | 63901s |
| | Ours (CiP) | 102s | 528s | 444s | 126s | 168s | 1654s |
| Memory | Vaze *et al.* (Vaze et al., 2022b) | 2206MB | 2207MB | 3760MB | 1354MB | 1394MB | 1902MB |
| | Ours (CiP) | 2535MB | 2932MB | 5848MB | 1392MB | 1451MB | 2205MB |

## 4.4 ABLATION STUDY

**Approaches to generate positive relations.** In Tab. 4, we compare our SNC with multiple different approaches to generate positive relations for joint contrastive learning, including directly using nearest neighbor (Zhong et al., 2021) in every mini-batch and conducting various clustering algorithms to obtain pseudo labels, *e.g.*, FINCH (Sarfraz et al., 2019), $k$-means (MacQueen et al., 1967), and semi-supervised $k$-means (Han et al., 2019; Vaze et al., 2022b). Non-hierarchical clustering methods ($k$-means and semi-supervised $k$-means) require a given cluster number. For $k$-means, we use the ground-truth class number. For semi-supervised $k$-means, we use both the ground truth and the overclustering number (twice the ground truth). We evaluate performance using both proposed SNC and semi-supervised $k$-means for comparison. It is clear that SNC reaches higher accuracy than semi-supervised $k$-means at test time. For generating pseudo positive relations, our method achieves best performance among all approaches. FINCH performs great on CIFAR-100 but degrades on CUB-200. We hypothesize that because FINCH is purely unsupervised without leveraging labelled data, it fails to generate reliable pseudo labels of more semantically similar instances on fine-grained CUB-200. Overclustering semi-supervised $k$-means achieves comparable performance on CUB-200 but performs bad on CIFAR-100. This might be caused by intrinsic poorer performance of semi-supervised $k$-means compared to proposed SNC, which results in worse pseudo labels. We further report the mean purity curve of pseudo labels generated by all clustering methods throughout training process in Fig. 3. We can observe that pseudo labels produced by SNC remain the highest purity on both datasets throughout the entire training process.

Table 4: **Results using different approaches to generate positive relations.** Semi-$k$-means$^\star$ denotes using semi-supervised $k$-means with an overclustering class number ($2 \times$ ground truth). The results evaluated with SNC are reported of normal size (left), and those evaluated with semi-supervised $k$-means are reported of smaller size (right).

| | CIFAR-100 | | | CUB-200 | | |
|---|---|---|---|---|---|---|
| Classes | All | Seen | Unseen | All | Seen | Unseen |
| w/ nearest neighbor | 80.4 74.9 | **82.9** 77.5 | 75.5 69.7 | 51.9 45.8 | 56.7 48.2 | 49.5 44.6 |
| w/ FINCH | 81.4 76.6 | 81.7 75.7 | **80.7** 78.6 | 51.4 47.9 | 51.8 45.1 | 51.3 49.3 |
| w/ k-means | 76.7 72.2 | 77.1 70.4 | 75.7 75.8 | 52.8 48.6 | 53.1 45.5 | 52.7 50.2 |
| w/ semi-k-means | 78.1 73.8 | 81.5 73.3 | 71.3 74.8 | 54.5 48.7 | 54.1 43.9 | 54.7 51.2 |
| w/ semi-k-means$^\star$ | 76.8 71.9 | 76.9 71.8 | 76.4 72.1 | 56.6 48.1 | 57.1 50.2 | **56.4** 47.1 |
| w/ SNC (ours) | **81.5** 76.5 | 82.4 75.1 | 79.7 79.3 | **57.1** 50.2 | **58.7** 48.8 | 55.6 51.0 |

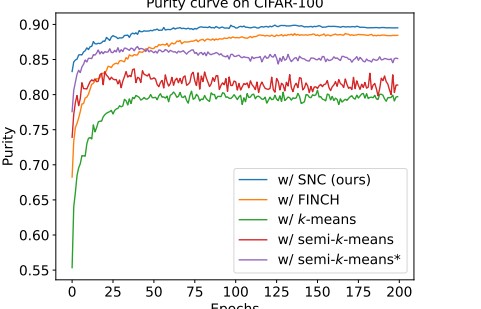
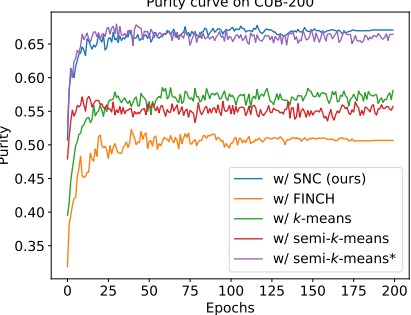

Figure 3: **Purity curve.**

**Effectiveness of cross-instance positive relations.** In this paper, we use SNC to generate pair-wise relations of unlabelled data, as well as relations between unlabelled and labelled data in supervised contrastive learning. In Tab. 5, we evaluate different settings to verify the effectiveness of both of these two relation types. We report evaluation results of SNC and semi-supervised $k$-means, showing higher accuracy achieved by SNC. Row (0) represents performance of the state-of-the-art GCD method of (Vaze et al., 2022b) without using any pseudo relations. Rows (1)-(3) show the effect of using different clustering methods to introduce relations of unlabelled and unlabelled pairs ($u$-$u$). All methods show improvements over (Vaze et al., 2022b). Among all relation generating methods, SNC brings the largest improvement, outperforming $k$-means and FINCH. Row (4) shows only adding pair-wise relations of labelled and unlabelled data ($u$-$\ell$) is not sufficient to boost baseline performance. Row (5) is our full method, which achieves the best performance. From (3)-(5), we clearly find fully using relations $u$-$u$ and $u$-$\ell$ generated from SNC benefits our method to the greatest extent, which also substantially improves performance on unseen categories.

Table 5: **Results using different relations.** $u$-$u$ denotes pair-wise relations between unlabelled and unlabelled data, and $u$-$\ell$ denotes pair-wise relations between unlabelled and labelled data. Rows (3)-(4) mean applying SNC on *all* data but only using $u$-$u$ or $u$-$\ell$ for pseudo positive relations.

| | $k$-means | FINCH | SNC | $u$-$u$ | $u$-$\ell$ | CIFAR-100 | | | CUB-200 | | |
|---|---|---|---|---|---|---|---|---|---|---|---|
| | | | | | | All | Seen | Unseen | All | Seen | Unseen |
| (0) | ✗ | ✗ | ✗ | ✗ | ✗ | 73.6 70.8 | 80.4 77.6 | 60.0 57.0 | 53.1 51.3 | 57.6 56.6 | 50.8 48.7 |
| (1) | ✓ | ✗ | ✗ | ✓ | ✗ | 77.2 73.1 | 78.3 74.4 | 74.9 70.7 | 56.0 51.0 | 53.8 42.8 | **57.1** 55.2 |
| (2) | ✗ | ✓ | ✗ | ✓ | ✗ | 80.3 78.2 | 79.5 76.9 | **81.7** 80.9 | 51.3 46.1 | 45.7 40.0 | 54.0 49.1 |
| (3) | ✗ | ✗ | ✓ | ✓ | ✗ | 80.5 76.5 | 80.6 76.3 | 80.3 76.9 | 56.6 52.7 | 57.2 51.5 | 56.3 53.3 |
| (4) | ✗ | ✗ | ✓ | ✗ | ✓ | 72.9 70.0 | 82.0 79.1 | 54.8 51.6 | 51.0 45.5 | 52.9 44.4 | 50.1 46.0 |
| (5) | ✗ | ✗ | ✓ | ✓ | ✓ | **81.5** 76.5 | **82.4** 75.1 | 79.7 79.3 | **57.1** 50.2 | **58.7** 48.8 | 55.6 51.0 |

## 5 CONCLUSION

We have presented a framework CiP for the challenging problem of GCD. Our framework leverages the cross-instance positive relations that are obtained with SNC, an efficient parameter-free hierarchical clustering algorithm we develop for the GCD setting. With the positive relations obtained by SNC, we can learn better representation for GCD, and the label assignment on the unlabelled data can be obtained from a single run of SNC, which is far more efficient than the semi-supervised $k$-means used in the state-of-the-art method. We also show that SNC can be used to estimate the unknown class number in the unlabelled data with higher efficiency.

## ETHICS STATEMENT

The potential negative impacts lie in two aspects. On the one hand, although the performance achieves the state-of-the-art, it still lags behind fully supervised models, making it risky to apply to scenarios with strict safety and accuracy requirements, *e.g.*, autonomous driving and medical image classification. On the other hand, due to unseen labels, manually checking the results is necessary in real applications, drawing attention to sensitive contexts (*e.g.*, private data) and inappropriate contents (*e.g.*, violent images).

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

# A    APPENDIX

## CONTENTS

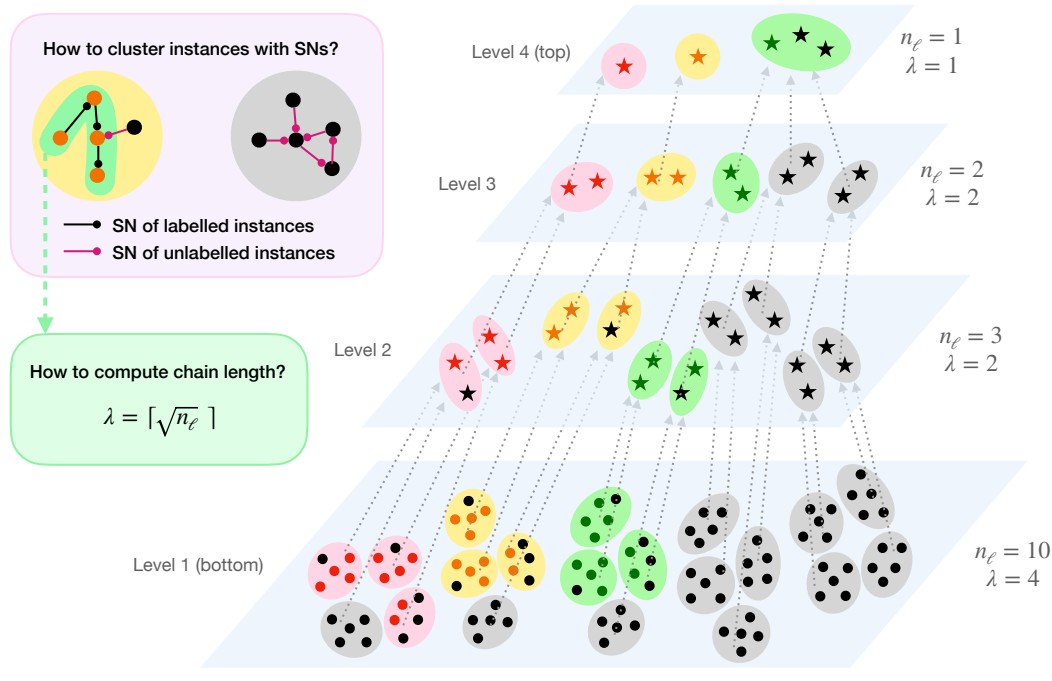

Figure 4: **A more detailed illustration of SNC.** SNC iteratively clusters instances from the bottom to the top, producing multiple levels of different partitions. At each level, the auto-adaptive chain length $\lambda$ is dynamically determined by the number of labelled 'instances' $n_\ell$ in each class. The connected components are extracted with selective neighbors (SNs), forming the clusters at each level.

## A.1 MORE ANALYSIS ON SNC

We present a more detailed illustration of our proposed SNC in Fig. 4. SNC is inspired by the idea from FINCH, but they are significantly different in two key aspects: (1) FINCH treats all instances the same and simply uses nearest neighbors to construct graphs; SNC uses a novel selective neighbor strategy tailored for the GCD setting to construct graphs, treating labelled and unlabelled instances differently. (2) SNC is able to cluster a mixed set of labelled and unlabelled data fully exploiting label supervision, but FINCH is not.

**Effectivenss of SNC on different learned features.** In Tab. 6, we evaluate SNC[1] on features extracted from DINO (Caron et al., 2021), GCD method of (Vaze et al., 2022b), and our method CiP. We also compare SNC with semi-supervised $k$-means (Han et al., 2019; Vaze et al., 2022b). We can observe that SNC surpasses semi-supervised $k$-means with a significant margin on all features, except those extracted by (Vaze et al., 2022b) on ImageNet-100. Moreover, semi-supervised $k$-means with our features performs better than with other features. Overall, SNC with our learned features gives the best performance.

Table 6: **Effectivenss of SNC on different learned features.**

| Clustering | Features | CIFAR-100 | | | ImageNet-100 | | | CUB-200 | | | SCars | | |
|---|---|---|---|---|---|---|---|---|---|---|---|---|---|
| | | All | Seen | Unseen | All | Seen | Unseen | All | Seen | Unseen | All | Seen | Unseen |
| Semi-$k$-means | DINO (Caron et al., 2021) | 60.4 | 63.1 | 54.9 | 72.8 | 70.6 | 73.8 | 36.7 | 37.9 | 36.0 | 12.3 | 13.7 | 11.6 |
| | Vaze et al. (Vaze et al., 2022b) | 74.5 | 81.9 | 60.0 | 69.2 | 66.6 | 70.5 | 53.5 | 59.9 | 50.3 | 40.8 | **67.6** | 27.8 |
| | CiP (ours) | 76.5 | 75.1 | 79.3 | 72.8 | 70.6 | 73.8 | 49.8 | 46.1 | 51.7 | 42.6 | 55.2 | 36.5 |
| SNC (ours) | DINO (Caron et al., 2021) | 65.5 | 69.0 | 58.3 | 76.8 | 81.1 | 74.6 | 36.7 | 35.0 | 37.5 | 12.4 | 15.8 | 10.7 |
| | Vaze et al. (Vaze et al., 2022b) | 77.8 | **87.4** | 58.6 | 61.4 | 76.7 | 53.8 | 55.9 | **61.6** | 53.0 | 41.3 | 62.9 | 30.8 |
| | CiP (ours) | **81.5** | 82.4 | **79.7** | **80.5** | **84.9** | **78.3** | **57.1** | 58.7 | **55.6** | **47.0** | 61.5 | **40.1** |

**Different choices of chain lengths $\lambda$.** The choice of chain lengths should be positively correlated to (but smaller than) the labelled instance number, while the number should not be too small. The

---

[1]When representing a clustering method here, SNC denotes selective neighbor clustering with one-by-one merging.

square root used in our paper is the simplest formulation we think of. In Tab. 7, we experiment on other formulations which satisfies the above relationship, *e.g.*, $\lambda = \lceil \sqrt[3]{n_\ell} \rceil$ and $\lambda = \lceil n_\ell/2 \rceil$, and our formulation performs the best. We also compare our dynamic $\lambda$ with a possible alternative of a fixed $\lambda$. For the fixed chain length, we conduct multiple experiments with different length values to find the best length giving the highest accuracy for each dataset. We observe that the best chain length varies from dataset to dataset, and there is no single fixed $\lambda$ that gives the best performance for all datasets. In contrast, our dynamic $\lambda$ consistently outperforms the fixed one, and it can automatically adjust the chain length for different datasets and different levels, without requiring any tuning nor validation like the fixed one.

Table 7: **Comparison of different formulations of chain length** $\lambda$. The best fixed length values are 8 for CIFAR-100 and 3 for CUB-200.

| | CIFAR-100 | | | CUB-200 | | |
|---|---|---|---|---|---|---|
| Classes | All | Seen | Unseen | All | Seen | Unseen |
| Fixed | 80.2 | 79.7 | **81.3** | 54.3 | **58.8** | 52.1 |
| $\lceil n_\ell/2 \rceil$ | 81.4 | **84.5** | 75.2 | 45.5 | 45.5 | 45.5 |
| $\lceil \sqrt[3]{n_\ell} \rceil$ | 72.5 | 77.1 | 63.2 | 42.4 | 45.0 | 41.1 |
| $\lceil \sqrt{n_\ell} \rceil$ (ours) | **81.5** | 82.4 | 79.7 | **57.1** | 58.7 | **55.6** |

**Impacts of different levels for positive relation generation.** A proper level for positive relation generation should overcluster the labelled data to some extent, such that reliable positive relations can be generated. Level 1 is not a valid choice because no positive relations can be generated if each instance is treated as a cluster. In Tab. 8, we present the performance using levels 2, 3, and 4 to generate pseudo labels and also compare with the previous state-of-the-art baseline by (Vaze et al., 2022b). We empirically find that the overclustering levels 3 and 4 are similarly good, while level 2 is worse because less positive relations are explored in each mini-batch. Even using level 2, our method still performs on par with (Vaze et al., 2022b).

Table 8: **Comparison of different levels.** Compare level 2, 3, 4 and baseline (Vaze et al., 2022b).

| | CIFAR-100 | | | CUB-200 | | |
|---|---|---|---|---|---|---|
| Classes | All | Seen | Unseen | All | Seen | Unseen |
| Baseline (Vaze et al., 2022b) | 70.8 | 77.6 | 57.0 | 51.3 | 56.6 | 48.7 |
| CiP w/ level 2 | 72.4 | 79.6 | 58.0 | 50.9 | 55.8 | 48.5 |
| CiP w/ level 3 (ours) | 81.5 | **82.4** | 79.7 | **57.1** | **58.7** | **55.6** |
| CiP w/ level 4 | **81.6** | 81.9 | **80.8** | 52.9 | 53.1 | 52.8 |

## A.2 A UNIFIED LOSS

In this paper, to leverage pseudo labels produced by SNC, we jointly train our model with two supervised contrastive losses, one using true positive relations of labelled data and the other using pseudo positive relations of all data. Indeed, it is possible to train the model with a unified loss by replacing the pseudo relations in the second term of our loss, and remove the first term. Formally, let $\mathcal{R}_{\mathcal{B}}(i)$ be the set of positive relations for instance $i$. The unified loss $\mathcal{L}_i^r$ can be written as

$$\mathcal{L}_i^r = -\frac{1}{|\mathcal{R}_{\mathcal{B}}(i)|} \sum_{q \in \mathcal{R}_{\mathcal{B}}(i)} \log \frac{\exp(\boldsymbol{z}_i \cdot \boldsymbol{z}_q/\tau)}{\sum_{n \in \mathcal{B}, n \neq i} \exp(\boldsymbol{z}_i \cdot \boldsymbol{z}_n/\tau)}, \tag{6}$$

where

$$\mathcal{R}_{\mathcal{B}}(i) = \begin{cases} \mathcal{G}_{\mathcal{B}}(i) \cup (\mathcal{P}_{\mathcal{B}}(i) \cap \mathcal{I}_{\mathcal{U}}) & \text{if } i \in \mathcal{I}_{\mathcal{L}} \\ \mathcal{P}_{\mathcal{B}}(i) & \text{if } i \in \mathcal{I}_{\mathcal{U}} \end{cases}, \tag{7}$$

$\mathcal{I}_{\mathcal{L}}$ and $\mathcal{I}_{\mathcal{U}}$ denote the instance indices of the labelled and unlabelled set respectively. In Tab. 9, we compare our two-term loss formulation with this unified loss formulation. It turns out that our two-term loss appears to be more effective. We hypothesize the performance degradation of Eq. (6) is caused by unbalanced granularity of labelled data and unlabelled data, due to mixture of overclustering pseudo labels and non-overclustering ground-truth labels.

Table 9: **Results using different loss formulations..**

| Classes | CIFAR-100 | | | CUB-200 | | |
|---|---|---|---|---|---|---|
| | All | Seen | Unseen | All | Seen | Unseen |
| Eq. (6) | 79.3 | 80.3 | 77.3 | 53.9 | 53.5 | 54.0 |
| Ours (CiP) | **81.5** | **82.4** | **79.7** | **57.1** | **58.7** | **55.6** |

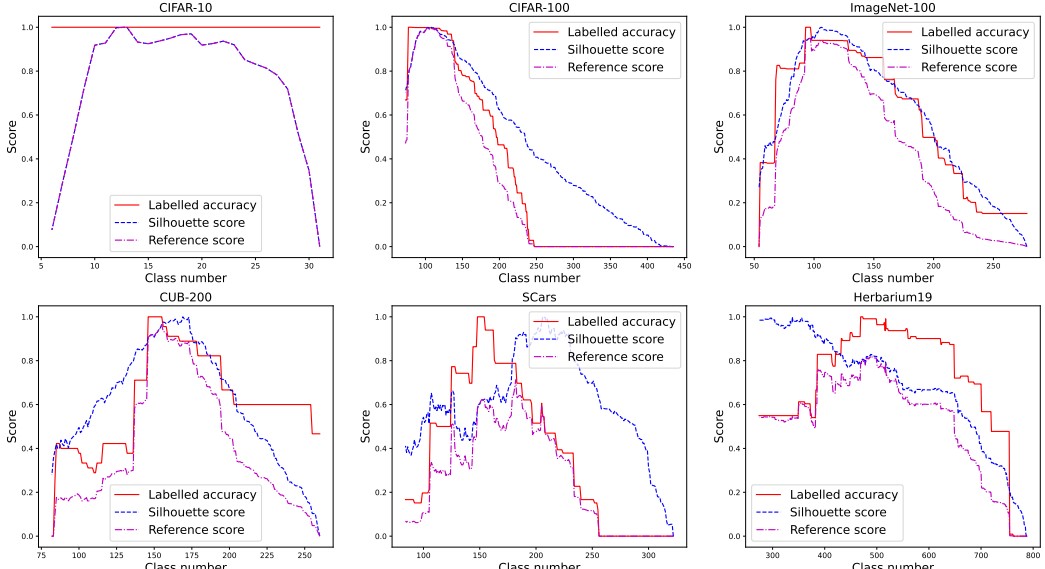

Figure 5: **Curves throughout class number estimation.** We report curves of accuracy on the labelled subset $\mathcal{D}_{\mathcal{L}}^v$, silhouette score on the unlabelled data $\mathcal{D}_{\mathcal{U}}$, and our reference score on $\mathcal{D}_{\mathcal{L}}^v \cup \mathcal{D}_{\mathcal{U}}$. Note that the $x$-axis should be read from right to left, as the merging start from the lower level to the upper level.

### A.3 CLASS NUMBER ESTIMATION

In Fig. 5, we show how labelled accuracy, silhouette score and reference score change throughout the whole procedure of class number estimation with one-by-one merging. The accuracy on the labelled instances or the silhouette score alone does not well fit the actual cluster number. By jointly considering both, we can see the actual class number aligns well with our suggested reference score.

### A.4 TIME EFFICIENCY

Here, we evaluate the time efficiency of CiP, including both category discovery and class number estimation.

**Category discovery efficiency.** The latency for the category discovery process mainly consist of two parts: feature extraction and label assignment. In Tab. 10, we present the feature extraction time. All methods consume roughly the same amount of time for feature extraction per image. RankStats+ (Han et al., 2021), UNO+ (Fini et al., 2021), and ORCA (Cao et al., 2022) assign labels with a linear classifier, thanks to the assumption of known category number. Hence, the label assignment process is simply done by a fast feed-forward pass of a linear classifier, costing omitable time ($< 0.0005$ second per image), though their performance lags. Our CiP and (Vaze et al., 2022b) contain the transfer clustering process for label assignment, for which CiP is 6-30 times faster than semi-supervised $k$-means used in (Vaze et al., 2022b) (see Tab. 11).

Table 11: **Time cost in clustering.**

| | CIFAR-10 | CIFAR-100 | ImageNet-100 | CUB-200 | SCars | Herbarium19 |
|---|---|---|---|---|---|---|
| Semi-$k$-means | 346s | 688s | 3863s | 256s | 356s | 6053s |
| Ours (SNC w/ one-by-one merging) | 58s | 111s | 118s | 36s | 50s | 917s |

Table 10: **Time cost in feature extraction per image.**

|  | Time cost |
|---|---|
| RankStats+ (Han et al., 2021) | 0.015s±0.001 |
| UNO+ (Fini et al., 2021) | 0.017s±0.001 |
| ORCA (Cao et al., 2022) | 0.015s±0.001 |
| Vaze *et al.* (Vaze et al., 2022b) | 0.014s±0.001 |
| Ours (CiP) | 0.014s±0.001 |

**Estimating class number.** Compared to repeatedly running $k$-means with different class numbers as in (Vaze et al., 2022b), CiP only requires a single run to obtain the estimated class number, thus significantly increasing efficiency. In Tab. 12, CiP is 40-150 times faster than (Vaze et al., 2022b), which utilizes $k$-means with the optimization of Brent's algorithm (Brent, 1971).

Table 12: **Time consumed in estimating class number.**

|  | CIFAR-10 | CIFAR-100 | ImageNet-100 | CUB-200 | SCars | Herbarium19 |
|---|---|---|---|---|---|---|
| Vaze *et al.* (Vaze et al., 2022b) | 15394s | 27755s | 64524s | 7197s | 8863s | 63901s |
| Ours (CiP) | 102s | 528s | 444s | 126s | 168s | 1654s |

## A.5 ATTENTION MAP VISUALIZATION

ViT (Dosovitskiy et al., 2020) has a multi-head attention design, with each head focusing on different context of the image. For the final block of ViT, the input $\mathbf{X} \in \mathbb{R}^{(HW+1)\times D}$, corresponding to a feature of $HW$ patches and a `[CLS]` token, is fed into multi-heads, which can be expressed as

$$MultiHead(\mathbf{X}) = [head_1, head_2, \ldots, head_h]\mathbf{W}^O \tag{8}$$

where

$$head_j = softmax(\frac{\mathbf{Q}_j\mathbf{K}_j^T}{\sqrt{d_k}})\mathbf{V}_j \tag{9}$$

$$\mathbf{Q}_j = \mathbf{X}\mathbf{W}_j^Q \tag{10}$$

$$\mathbf{K}_j = \mathbf{X}\mathbf{W}_j^K \tag{11}$$

$$\mathbf{V}_j = \mathbf{X}\mathbf{W}_j^V \tag{12}$$

where $d_k$ is the dimension of queries and keys. In our model, patch size is $16 \times 16$ pixels and $HW = 14 \times 14 = 196$. The number of heads $h$ is 12. Referring to (Vaswani et al., 2017), consider attention map of head $j$ $\mathbf{A}_j = softmax(\frac{\mathbf{Q}_j\mathbf{K}_j^T}{\sqrt{d_k}}) \in [0,1]^{(HW+1)\times(HW+1)}$. $\mathbf{A}_j$ describes the similarity of one feature to every other feature captured in head $j$. The first row of $\mathbf{A}_j$ shows how head $j$ attends `[CLS]` token to every spatial patch of the input image. In Fig. 6, we visualize some of the interpretable attention heads to show semantic regions that ViT attends to. We can observe that our model CiP, as well as DINO (Caron et al., 2021) and (Vaze et al., 2022b), can attend to specific semantic object regions. For instance, CiP attends three heads respectively to 'license plate', 'light' and 'wheels' for Stanford Cars (head 1 fails in row 1), and to 'body', 'head' and 'neck' for CUB-200.

## A.6 DATA SPLITS

In Tab. 13, we show the details on data splits of CIFAR-10 (Krizhevsky et al., 2009), CIFAR-100 (Krizhevsky et al., 2009), ImageNet-100 (Deng et al., 2009), CUB-200 (Wah et al., 2011), Stanford Cars (Krause et al., 2013) and Herbarium19 (Tan et al., 2019) in our experiments.

## A.7 SPECIAL CASES OF UNLABELLED DATA

In the real world, we may meet the scenarios where unlabelled data are all from seen or unseen classes. We investigate into such scenarios and conduct experiments to validate effectiveness of our method. Our experiment are under two settings: (1) applying our pretrained models in the main paper to seen-only and unseen-only unlabelled data; (2) retraining the models with seen-only and unseen-only unlabelled data. In Tab. 14, we can observe that our model maintains strong performance in all cases.

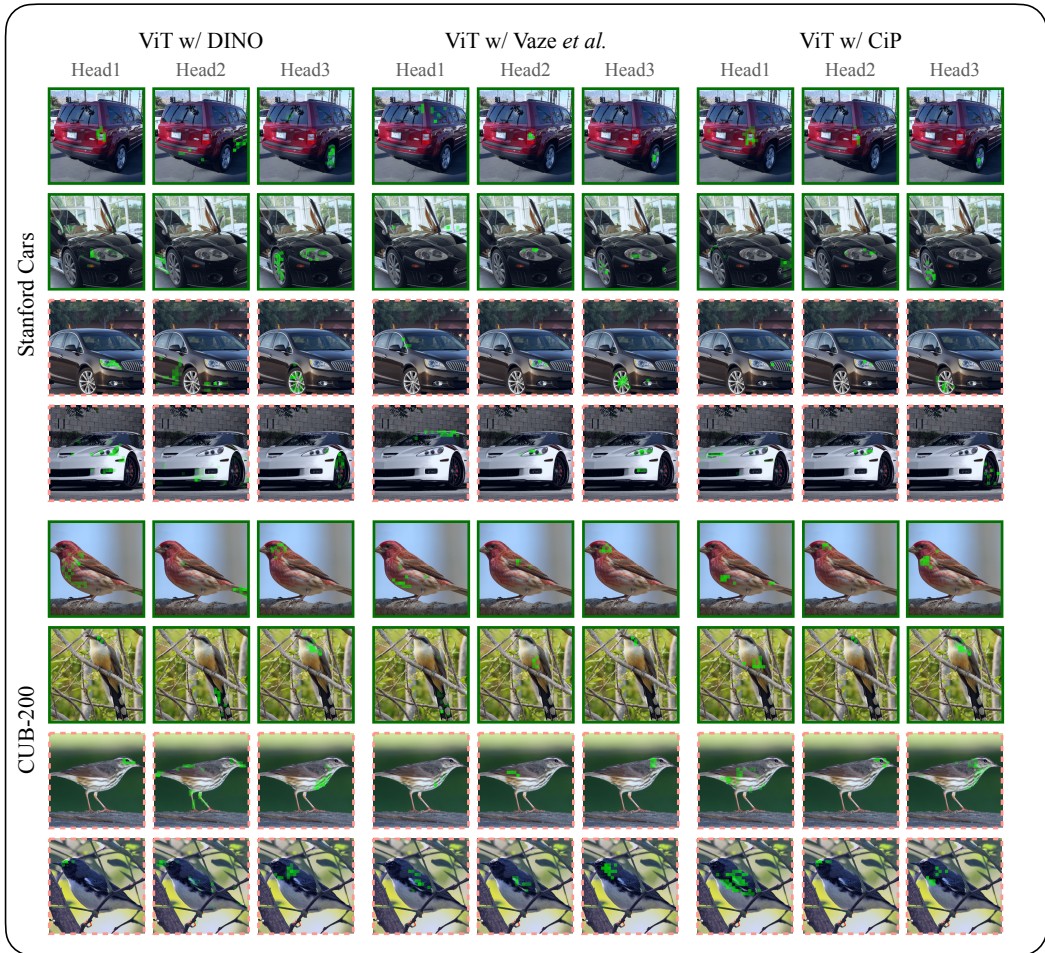

Figure 6: **Attention visualizations.** We report visualization results of DINO (Caron et al., 2021) (left), (Vaze et al., 2022b) (middle) and CiP (right) on Stanford Cars (top) and CUB-200 (bottom). For each dataset, we show two rows of 'Seen' categories (solid green box) and two rows of 'Unseen' categories (dashed red box). Zoom in to see attention details.

Table 13: **Data splits of all datasets.** We present the number of classes in the labelled and unlabelled set ($|\mathcal{Y}_\mathcal{L}|$, $|\mathcal{Y}_\mathcal{U}|$), and the number of images ($|\mathcal{D}_\mathcal{L}|$, $|\mathcal{D}_\mathcal{U}|$).

|  | CIFAR-10 | CIFAR-100 | ImageNet-100 | CUB-200 | SCars | Herbarium19 |
|---|---|---|---|---|---|---|
| $|\mathcal{Y}_\mathcal{L}|$ | 5 | 80 | 50 | 100 | 98 | 341 |
| $|\mathcal{Y}_\mathcal{U}|$ | 10 | 100 | 100 | 200 | 196 | 683 |
| $|\mathcal{D}_\mathcal{L}|$ | 12.5k | 20k | 32.5k | 1.5k | 2.0k | 8.5k |
| $|\mathcal{D}_\mathcal{U}|$ | 37.5k | 30k | 97.5k | 4.5k | 6.1k | 25.7k |

Table 14: **Performance on seen-only and unseen-only unlabelled data.** "original setting" denotes the performance of CiP dealing with GCD; "direct testing" denotes the performance of CiP dealing with seen-only or unseen-only unlabelled data using pretrained GCD model; "retraining" denotes the performance of retrained CiP dealing with seen-only or unseen-only unlabelled data.

|  |  | CIFAR-10 | CIFAR-100 | ImageNet-100 | CUB-200 | SCars | Herbarium19 |
|---|---|---|---|---|---|---|---|
| Seen | original setting | 97.5 | 82.4 | 84.9 | 58.7 | 61.5 | 45.4 |
|  | direct testing | 98.5 | 84.4 | 83.3 | 79.1 | 72.0 | 55.4 |
|  | retraining | 98.9 | 87.0 | 87.3 | 81.9 | 75.2 | 66.3 |
| Unseen | original setting | 97.7 | 79.7 | 78.3 | 55.6 | 40.1 | 32.6 |
|  | direct testing | 97.6 | 82.7 | 74.3 | 56.5 | 39.3 | 37.9 |
|  | retraining | 98.4 | 78.9 | 79.3 | 60.4 | 42.5 | 41.3 |

### A.8 LIMITATIONS

We note limitations of our method. In our current experiments, we consider images from the same curated dataset. However, in practice, we might want to transfer concepts from one dataset to another, which may have different data distribution, introducing more challenges. For example, the unlabelled data could follow the long-tailed distribution. Another limitation is that currently, we need to train the model on both labelled and unlabelled data jointly. However, in real world, there are often cases in which we do not have access to any labelled data from the seen classes when facing the unlabelled data. We consider these as our future research directions.

### A.9 LICENSE FOR EXPERIMENTAL DATASETS

All datasets used in this paper are permitted for research use. CIFAR-10 and CIFAR-100 (Krizhevsky et al., 2009) are released under MIT License, allowing for research propose. ImageNet-100 is the subset of ImageNet (Deng et al., 2009), which allows non-commercial research use. Similarly, CUB-200 (Wah et al., 2011), Stanford Cars (Krause et al., 2013) and Herbarium19 (Tan et al., 2019) are also exclusive for non-commercial research purpose.

