# OpenReview forum: "Effective Cross-instance Positive Relations for Generalized Category Discovery"
_ICLR.cc/2023/Conference — Submitted to ICLR 2023_

### Official Review · Reviewer_jVVh · 2022-10-21

**Confidence:** 4
**Correctness:** 3
**Technical Novelty And Significance:** 2
**Empirical Novelty And Significance:** 3
**Recommendation:** 5

**Clarity, Quality, Novelty And Reproducibility:**

The overall quality is good. The paper is clearly written and the proposed method is novel.

**Strength And Weaknesses:**

Strength:

- The paper is well organized and easy to follow.
- The proposed method is new and demonstrated to be effective.

Weaknesses:

- The propsed method is a simple extension of the existing hirarchical clustering algorithm, with intuitive modification that is not properly justified theoretically. For instance, why should samples linked by chains? Why should chains be connected under certain regulations? Why is it significant and necessary? These are just proposed but not well justified.


**Summary Of The Paper:**

This paper tackles the task of generalized class discovery where both labelled and unlabelled data can be encountered during inference. The authors proposed a semi-supervised hierarchical clustering algorithm to solve this problem, which is built upon a known clustering method. The proposed approach is equipped with the capability of generating reliable pseudo labels as well as fast estimating unseen class numbers. The proposed method is evaluated on several popular benchmarks, which demonstrates promising results.

**Summary Of The Review:**

A well written paper with less theoretical contribution.

---

> ### Author Response · Authors · 2022-11-08
> **Answer to Reviewer jVVh**
>
> Thank you for your comments.  We address the concern as follows:
>
> > The proposed method is a simple extension of the existing hierarchical clustering algorithm, with intuitive modification that is not properly justified theoretically. For instance, why should samples be linked by chains? Why should chains be connected under certain regulations? Why is it significant and necessary? These are just proposed but not well justified.
>
> We discussed the motivation for using chains in lines 1-28 in the third paragraph of Sec. 3.3 and the details in lines 28-33 in the third paragraph of Sec. 3.3. In brief, we use chains to **avoid all instances being connected together forming a single cluster**, leading to the degenerated solution. We carefully considered the data property of the GCD problem when constructing the chain to best fit the problem.
>
> To this end, we propose certain regulations. First, we *dynamically* decide the chain length at different hierarchies. This value is positively correlated to (but smaller than) the labelled instance number because of the intuition that the more labelled instances, the more instances we hope to include in one cluster (ie, the longer chain). Second, each labelled instance in a chain can only be the selective neighbor of another labelled instance *once*. If not, labelled data would not be connected by chains. Instead, labelled data will be connected by many interlaced branches, which tend to connect all labelled data together unexpectedly. Third, the selective neighbor of an unlabelled instance can be a labelled or an unlabelled instance, depending on its actual distances to other instances. This is because we do *not* constrain unlabelled data in clustering and just let them connect to their nearest neighbor. These regulations are critical for GCD setting to generate reliable pseudo labels and have not been considered by existing works.
>
> Finally, the experimental results also validated the effectiveness of our method.

---

### Official Review · Reviewer_T62F · 2022-10-23

**Confidence:** 5
**Correctness:** 3
**Technical Novelty And Significance:** 3
**Empirical Novelty And Significance:** 3
**Recommendation:** 5

**Clarity, Quality, Novelty And Reproducibility:**

Clarity and Quality: The overall writing of this paper is good. However, the Sec3.3 and 3.4 are hard to understand.

Novelty: The proposed clustering and learning framework is not very novel.

Reproducibility: Algorithms are provided in the paper. However, the proposed method is quick complex. I think it is not easy to reproduce the method without the source code.

**Strength And Weaknesses:**

Strength

+ This paper studies a new problem and practical problem in the community.

+ This paper is well written and easy to follow.

+ An effective approach is proposed for GCD, which largely improves the sota performance.

+ Extensive experiments are provided to show the benefit of the proposed method.



Weaknesses

- Despite the high performance, the novelty of the proposed method is somewhat limited. Both clustering-based positive pair mining and supervised contrastive learning based on the selected pairs are not new techniques in the community.

- The description of Sec.3.3 and Sec.3.4 is hard to understand.

- The improvement over the original FINCH sometimes is limited. For example, in Table 5, when applying SNC on only unlabeled data, it should be mostly similar to FINCH with unlabeled data. However, the full method does not produce a clear improvement on the SNC with u-u.

- The memory cost comparison between Cip and GCD should be provided in Table 3.




**Summary Of The Paper:**

This paper considers the problem of generalized category discovery (GCD). To solve this problem, the authors propose a positive-pair mining approach for facilitating the contrastive learning, which encourages the network learn more discriminative representation. Extensive experiments on several GCD datasets show the benefit of the proposed method.

**Summary Of The Review:**

This paper studies a new task and proposes an effective method for this task, which achieves new SOTA performance. However, the novelty of the proposed framework is somewhat limited. In addition, compared to the original FINCH method, the improvement of the proposed method is sometime limited.

---

> ### Author Response · Authors · 2022-11-08
> **Answer to Reviewer T62F**
>
> Thank you for your comments. We address the concerns as follows:
>
> > Despite the high performance, the novelty of the proposed method is somewhat limited. Both clustering-based positive pair mining and supervised contrastive learning based on the selected pairs are not new techniques in the community.
>
> We agree that constrative learning has been adopted in the NCD and GCD literature for representation. For the GCD problem, the work (Vaze et al., 2022) is the first to adopt self-supervised contrastive learning and supervised contrastive learning on the mixed set of unlabelled and labelled data. However, the labelled and unlabelled data are considered separately and the connections among labelled/unlabelled data and seen/unseen classes are not considered, not fully exploiting the property of the data for representation learning on GCD and leading to the risk of biased representation. To our knowledge,  our CiP is the first to extend supervised contrastive learning to *all* data points in GCD by generating pseudo-positive relations across all instances through SNC, alleviating the bias on the labelled data, which is a common issue of other NCD/GCD methods, and allowing unbiased representation learning and label assignment without knowing the cluster number. Moreover, we propose a more efficient class number estimation method with a novel joint reference score, faster than (Vaze et al., 2022) by 40-150 times.
>
> > The description of Sec.3.3 and Sec.3.4 is hard to understand.
>
> Sec. 3.3 and Sec 3.4 contain the key information describing the detailed design of SNC and how it works for **pseudo-label generation**, **class label assignment**, and **class number estimation**. Due to the page limit, we try to make it as succinct as possible. We would like to refer the reviewer to Appendix A.1 for more illustrations with figures. If there is still anything unclear, we would be happy to expand.
>
> > The improvement over the original FINCH sometimes is limited. For example, in Table 5, when applying SNC on only unlabeled data, it should be mostly similar to FINCH with unlabeled data. However, the full method does not produce a clear improvement on the SNC with u-u.
>
> Indeed, SNC with $u$-$u$ (row (3) in Table 5) does not mean “applying SNC on only unlabeled data”, and SNC with $u$-$u$ is significantly different from FINCH with $u$-$u$. SNC with $u$-$u$ (row (3) in Table 5) means applying SNC on **all data** (the whole set of mixed labelled and unlabelled data) but only using positive relations among unlabelled data for pseudo labels (and dropping other types of relations). In contrast, FINCH with $u$-$u$ (row (2) in Table 5) means applying FINCH on **only unlabelled data** and using positive relations among unlabelled data for pseudo labels. Table 5 reports the comparison using different relations in our framework. We can observe that the full method (using SNC-produced $u$-$u$ and $u$-$l$ relations) outperforms all the others, especially on fined-grained CUB-200 with quite a large margin. We have added more descriptions to the caption of Table 5 to avoid misunderstandings.
>
> In Table 4, we report the direct comparison of different pseudo-label generators by replacing SNC with different alternatives (including FINCH) in our framework. As can be seen, SNC outperforms all the others consistently.
>
> In addition,  to further address this concern, we compare the performance on CIFAR10 and S-Cars using SNC and FINCH as the pseudo-label generator on **all data** and the results are:
>
> | CIFAR10 | All | Seen | Unseen|
> |-|-|-|-|
> | w/ FINCH | 97.4 | 96.5  | **97.8** |
> | w/ SNC | **97.7** | **97.5** | 97.7 |
>
> | S-Cars | All | Seen | Unseen|
> |-|-|-|-|
> | w/ FINCH | 42.7 | 54.6  | 37.0 |
> | w/ SNC | **47.0** | **61.5** | **40.1** |
>
> Similar to the performance on CIFAR-100 and CUB-200 reported in Table 4, SNC consistently outperforms FINCH, especially on fine-grained datasets. This result further validates that utilizing label information by SNC is essential when feature discrimination is deficient in fine-grained cases.
>
> > The memory cost comparison between Cip and GCD should be provided in Table 3.
>
> We follow the suggestion to measure the memory cost between CiP and (Vaze et al., 2022). The results are shown below. We have added the memory cost comparison to the paper.
>
> | Mem (MB) | CIFAR10 | CIFAR100 | ImgNet100 | CUB200 | SCars | Herbarium19 |
> |-|-|-|-|-|-|-|
> | Vaze et al. | 2206 | 2207 | 3760 | 1354 | 1394 | 1902 |
> | Ours (CiP) | 2535 | 2932 | 5848 | 1392 | 1451 | 2205 |
>
> Our method costs slightly more memory than (Vaze et al., 2022), mainly due to the hierarchical graph construction. However, our method is 40-150 times faster.
>
> **For the novelty concern**, we have further clarified our contributions in the [general comment](https://openreview.net/forum?id=hag85Gdq_RA&noteId=vsqiAVcffC).
> **For reproducibility**, we will release the source code after acceptance.
>
> [Reference]
> Vaze et al., Generalized category discovery. CVPR 2022.

---

### Official Review · Reviewer_Kvso · 2022-10-24

**Confidence:** 3
**Correctness:** 3
**Technical Novelty And Significance:** 3
**Empirical Novelty And Significance:** 3
**Recommendation:** 6

**Clarity, Quality, Novelty And Reproducibility:**

The originality should be better clarified.


**Strength And Weaknesses:**

Strength
1. I idea using both labeled and unlabelled data is intuitive.
2. The performance improvement is quite impressive, e.g., up to 10% accuracy improvement in some settings.
3. The paper is well-written and easy to follow.

Weaknesses
1. It seems the key contribution is the SNC which can help generate pseudo labels. The authors claim that SNC is inspired by and similar to FINCH. However, the difference with FINCH is not well clarified.
2. The authors use SNC to generate pseudo labels. I wonder how to ensure the accuracy of generated labels. It may hurt model training if generating many wrong labels.

**Summary Of The Paper:**

This work proposes cross-instance positive relations that introduce a selective neighbor clustering module to help generate pseudo labels for contrastive learning. It can promote using both the labeled and unlabelled data to help more representative representation. It presents experiments on several benchmarks to demonstrate its performance.


**Summary Of The Review:**

I think the overall idea is OK and the experiments are quite convincing. The contribution should be better clarified.

---

> ### Author Response · Authors · 2022-11-08
> **Answer to Reviewer Kvso**
>
> Thank you for your comments. We address the concerns as follows:
> > It seems the key contribution is the SNC which can help generate pseudo labels. The authors claim that SNC is inspired by and similar to FINCH. However, the difference with FINCH is not well clarified.
>
> SNC is inspired by the idea from FINCH, but they are different in two key aspects:
> 1. FINCH treats all instances the same and simply uses nearest neighbors to construct graphs; SNC uses a novel selective neighbor strategy tailored for the GCD setting to construct graphs, treating labelled and unlabelled instances differently. (See Sec. 3.3 for the full description)
>
> 2. SNC is able to cluster a mixed set of labelled and unlabelled data fully exploiting label supervision, but FINCH is not.
>
> Due to the page limit, we added this difference clarification in Appendix A.1.
>
> > The authors use SNC to generate pseudo labels. I wonder how to ensure the accuracy of generated labels. It may hurt model training if generating many wrong labels.
>
> We agree that many wrong labels are harmful. We use a pretrained DINO ViT model with self-supervision as initialization, which can produce discriminative representations, thus ensuring the clustering quality at the beginning. The training of CiP is a self-adaptive process in which the cluster purity keeps growing, leading to reduced wrong pseudo labels over time (see Figure 3). Because we use pair-wise pseudo labels, which benefit from clusters with high purity, achieved by the over-clustering of SNC. Thus, the quality of the pseudo labels can be measured by purity as shown in Figure 3.

---

### Author Response · Authors · 2022-11-08
**General comment**

We thank all reviewers for their recognition of our work: (1) **a well-written paper** [reviewers Kvso, T62F, jVVh] (2) **a new and practical problem** [reviewer T62F] (3) **an intuitive idea** [reviewer Kvso] (4) **a novel** [reviewers jVVh] **and effective** [reviewers T62F, jVVh] **approach** (5) **extensive experiments** [reviewer T62F] (6) **impressive performance** [reviewers Kvso, T62F, jVVh].

We would like to clarify that our contributions are not simply proposing a clustering method. We tackle a very new and challenging problem, namely generalized category discovery (GCD), including extra challenges than pure clustering: (1) learning unbiased representation on unlabelled data contains instances from seen or unseen classes; (2) the class number of unlabelled data is not known; (3) label assignment on the unlabelled data.

To summarize the contributions of this paper:
- we propose a new GCD framework, CiP, to achieve joint supervised contrastive learning on labelled and unlabelled data using pseudo-positive relations.
- we propose a novel clustering method SNC, which is suitable for the GCD problem. SNC can not only generate reliable pseudo labels but also assign classes with a given class number.
- we propose a more efficient class number estimation method with a novel joint reference score based on SNC. Time efficiency is 40-150 times higher with negligible memory cost increment and comparable performance.

The SOTA method (Vaze et al., 2022) only uses self-supervised learning to learn from different augmentations of unlabelled data. It fails to consider relationships between labelled and unlabelled data, thus causing learning bias to the labelled set. Our paper is the first work to exploit labelled-unlabelled relations and unlabelled-unlabelled relations for unbiased representation learning for GCD. We propose SNC to produce reliable cross-instance relations for labelled and unlabelled data by carefully considering the property of GCD.
The produced relations make it possible to employ contrastive learning on unlabelled data in a “*supervised*” manner.

Though one can use off-the-shelf clustering methods to produce pseudo labels, no existing method meets the following properties of SNC simultaneously, which are critical to GCD: (1) utilizing **label supervision** to help cluster unlabelled data; (2) **not requiring the cluster number** to be known; (3) **high-purity over-clustering** for reliable pair-wise pseudo labels for unbiased representation learning.
Experiments have validated that SNC leads to the best performance of GCD among all other possible options of pseudo-label generation methods (see Table 4).

We also propose a much more efficient approach to class number estimation. So far, there has been only one method (Vaze et al., 2022) considering class number estimation in GCD. We propose an SNC-based estimation method with a novel joint reference score, which is 40-150 times faster than the method of (Vaze et al., 2022) with comparable new class estimation accuracy.

[Reference]
Vaze et al., Generalized category discovery. CVPR 2022.

---

### Author Response · Authors · 2022-11-10
**A gentle reminder**

Dear Reviewers,

We thank you for the valuable comments. We have carefully addressed all the concerns and revised our paper accordingly. If there is any point that you feel is unclear or any remaining concern, please let us know. We will be happy to address any further concerns.

Best regards,
Authors of Paper1008

---

### Decision · Program_Chairs · 2023-01-20

**Decision:**

Reject

**Justification For Why Not Higher Score:**

Although the experiments are rich, the novelty is limited, which makes this paper difficult to reach the bar of ICLR.

**Justification For Why Not Lower Score:**

N/A

**Metareview: Summary, Strengths And Weaknesses:**

Based on the collected information from all reviewers and my personal judgment, I can make the initial recommendation on this paper, ** reject**. Here are the comments that I summarized, which include my opinion and evidence.

**Research Problem**

The authors study the open-world clustering problem, where the dataset is partially labeled and the unlabeled data contain instances from both seen and unseen categories.

**Motivation**

The motivations come from the drawback of one state-of-the-art algorithm (Vaze et al., 2022b). (1) Vaze’s method considers labeled and unlabeled data independently, (2) Vaze’s method requires a known category number, and (3) Vaze’s method is inefficient. The second point is not true. Vaze’s method can also estimate the unknown category number for the unlabeled data based on a semi-supervised K-means clustering. For the third point, there exist several estimation methods, which do not need multiple runs. The authors constrain their scope into a small range. For the first point, the motivations are not strong enough and make this paper an incremental work.

**Philosophy**

For the first motivation, the authors aim to explore all possible interactions between labeled and unlabeled instances. For the unlabeled data, pseudo labels from their designed clustering algorithms are used to generate cross-instance positive relations. This is very straightforward. Moreover, it might suffer from self-bias due to no supervision from unlabeled data. Such pseudo labels might further enforce the cross-instance positive relations derived from the pseudo labels.

**Presentation**

In general, this paper is easy to follow. One minor issue: I suggest giving a definition of the cross-instance positive relations.

**Technique**

Several reviewers and I have concerns about the technical novelty. Most techniques are built on existing frameworks. For the new content in Section 3, the novelty is limited and incremental compared to the existing literature.

**Experiments**

The biggest advantage of this paper is that the experiments are extensive and solid. Very nice.

No objection was raised from the reviewers on the rejection recommendation.

**Summary Of Ac-Reviewer Meeting:**

This is not a borderline paper.